# Synthesis, Characterization and Biological Evaluation of Benzothiazole–Isoquinoline Derivative

**DOI:** 10.3390/molecules27249062

**Published:** 2022-12-19

**Authors:** Weihua Liu, Donghai Zhao, Zhiwen He, Yiming Hu, Yuxia Zhu, Lingjian Zhang, Lianhai Jin, Liping Guan, Sihong Wang

**Affiliations:** 1Food and Pharmacy College, Zhejiang Ocean University, Zhoushan 316022, China; 2Pharmacy College, Jilin Medical University, Jilin 132013, China; 3Key Laboratory of Natural Resource of the Changbai Mountain and Functiaonal Molecules, Ministry of Education, Yanbian University, Yanji 133000, China

**Keywords:** neurodegenerative disease, antidepressant, benzothiazole–isoquinoline derivatives, MAO–B, BuChE, molecular docking

## Abstract

Currently, no suitable clinical drugs are available for patients with neurodegenerative diseases complicated by depression. Based on a fusion technique to create effective multi–target–directed ligands (MTDLs), we synthesized a series of (*R*)–*N*–(benzo[d]thiazol–2–yl)–2–(1–phenyl–3,4–dihydroisoquinolin–2(1*H*)–yl) acetamides with substituted benzothiazoles and (*S)*–1–phenyl–1,2,3,4–tetrahydroisoquinoline. All compounds were tested for their inhibitory potency against monoamine oxidase (MAO) and cholinesterase (ChE) by in vitro enzyme activity assays, and further tested for their specific inhibitory potency against monoamine oxidase B (MAO–B) and butyrylcholinesterase (BuChE). Among them, six compounds (**4b**–**4d**, **4f**, **4g** and **4i**) displayed excellent activity. The classical antidepressant forced swim test (FST) was used to verify the in vitro results, revealing that six compounds reduced the immobility time significantly, especially compound **4g**. The cytotoxicity of the compounds was assessed by the MTT method and Acridine Orange (AO) staining, with cell viability found to be above 90% at effective compound concentrations, and not toxic to L929 cells reversibility, kinetics and molecular docking studies were also performed using compound **4g**, which showed the highest MAO–B and BuChE inhibitory activities. The results of these studies showed that compound **4g** binds to the primary interaction sites of both enzymes and has good blood–brain barrier (BBB) penetration. This study provides new strategies for future research on neurodegenerative diseases complicated by depression.

## 1. Introduction

Neurodegenerative diseases represent an important health issue because of the aging population [1]. Alzheimer’s disease (AD) accounts for the largest proportion of people suffering from neurodegenerative disorders, followed by Parkinson’s disease (PD) [2,3,4]. AD is rapidly becoming an economic burden to nations, with mortality rates associated with AD increasing [5,6,7]. In addition, depressive symptoms are prevalent in individuals with neurodegenerative diseases, and according to recent meta–analysis research, depression is present in roughly 40% of AD cases [8], and can be fatal [9].

Notably, depression may be a risk factor associated with AD development, and neurodegeneration in the AD brain may also predispose individuals to depression [10,11]. AD and depression have distinct neuropathological features. Nonetheless, there are some similarities in the mechanisms (e.g., dysregulation of monoaminergic neurotransmitters, decreased nutritional support, HPA axis disruption, neuroinflammation, excitotoxicity and oxidative damage) that may play a role in the neurodegenerative processes caused by two illnesses [12,13,14,15,16,17,18,19,20]. The use of antidepressants in AD is linked with severe side effects such as hyponatremia [21], cardiotoxicity [22] and an increase in possible hemorrhaging [23,24,25]. In one meta–analysis, treatment of neurodegenerative diseases, especially AD comorbid depression included selective serotonin reuptake inhibitors (SSRIs), noradrenergic and specific serotonergic antidepressants, as well as reversible selective monoamine oxidase inhibitors. These drugs have also been studied without significant treatment effects [26,27,28].

The use of antidepressants to treat neurodegenerative diseases has not shown substantial therapeutic benefit, and additional research on the disease pathways and pathophysiology of neurodegenerative diseases complicated by depression is required to elucidate the pathogenesis and identify the preventive treatment of these diseases [29,30,31,32]. Dual–drug or drug–combination techniques and enhancement strategies are likely to be effective in the combined treatment of these diseases. Currently, therapy to treat neurodegenerative diseases for patients with comorbid depression remains an urgent, unmet global issue. Unfortunately, no clinically authorized medicine that prevents or reduces the progression of the illness has been identified.

The goal of this study was to develop compounds that cure symptoms of depression while also preventing neurodegeneration, thereby confirming the validity of our medicinal design concept. In this report, we used a fusion technique to create effective multi–target–directed ligands (MTDLs) [33,34,35] and combined this approach with previous research findings [36,37,38,39], as shown in Figure 1. Benzothiazole (fragment A) is a heterocycle often used in drug discovery research [40,41]. With various benzothiazole–bearing compounds displaying potential therapeutic effects against AD and depression, this pharmacophoric category is essential for drug development against the neurodegenerative disease comorbid depression sector [42,43,44,45]. Sabeluzole, a benzothiazole–based agent, has been demonstrated to slow the clinical course of AD [46]. In addition, isoquinoline (fragment C) has a diverse set of biological activities, including antidepressant, anti–AD, anti–PD and other neuroprotective activities [47,48,49,50,51]. Benzothiazole (fragment A) and isoquinoline (fragment C) are connected by an amide bond (fragment B) because amide compounds are used to treat illnesses of the central nervous system, such as anxiety, schizophrenia, epilepsy and depression, and have antioxidant pharmacological activity [52,53,54,55]. The most relevant parameters of the newly created MTDLs include physicochemical features of the compounds in connection to their drug–likeness and success in the drug development process. The bulk of the compounds have met Lipinski’s rule of five criteria and bioavailability requirements [56]. In conclusion, the designed and synthesized series of (*R*)–*N*–(benzo[d]thiazol–2–yl)–2–(1–phenyl–3,4–dihydroisoquinoline–2(1*H*)–yl)acetamides have good potential (Figure 1). The compounds can be a potential therapy for neurodegenerative disease and depression comorbidities and as useful reference prototypes for the developing new treatment strategies.

## 2. Results and Discussion

### 2.1. Chemistry

#### Synthetic Route and Method of Novel Benzothiazole–Isoquinoline Derivatives

The synthetic methods used to obtain the target derivatives are presented in Figure 1. The structures of the new derivatives were verified by HPLC, IR, ^1^H NMR, ^13^C NMR and mass spectrometry analyses.

In the IR spectra of **4a**–**4p** derivatives, an NH stretching band at 3289–3261 cm^−1^ for the single bond was observed, and a CO band for the amide double bond (1619–1512 cm^−1^) appeared, which supports target amidation. In the ^1^H NMR spectrum, –NH proton signals were observed at 9.15–10.35 ppm. The α–CH_2_ proton signals of the amides of **4a**–**4p** were observed at 3.95–4.79 ppm, and the C_2_–H protons in 1–phenyl–3,4–dihydro–isoquinolin–2(1*H*)–yl were observed to resonate at 4.72–6.79 ppm. In addition, the ^13^C NMR spectra of **4a**–**4p** also confirmed the presence of amide bonds, benzothiazole groups and 1–phenyl–3,4–dihydroisoquinolin groups. The experimental section contains additional synthetic pathways and spectroscopic details.

### 2.2. In Vitro Biological Evaluation

#### 2.2.1. Inhibitory Activity of Derivatives on MAO

Derivatives (**4a**–**4p**) were tested for MAO inhibitory activity, and pargyline, rasagiline and clorgyline were used as reference compounds. Table 1 displays the results. We discovered that the majority of benzothiazole–isoquinoline derivatives have excellent MAO inhibitory activity. Among them, analog **4g** (IC_50_ = 14.80 ± 5.45 μM) had the strongest antagonistic effect on MAO activity, and analog **4d** (IC_50_ = 64.83 ± 4.20 μM) had the highest inhibitory rate on MAO (43.37%).

The results of the structure–activity relationship (SAR) for the benzothiazole–isoquinoline derivatives on MAO are shown in Table 1. The results for compound **4g** (57.11%, IC_50_ = 14.80 ± 5.45 μM) revealed that introducing an *o*–CH_3_ group on the benzothiazole ring increased MAO inhibition. In contrast, a sharp decrease in MAO inhibition was observed for compound **4h** (16.72%, IC_50_ = 76.37 ± 2.58 μM) and **4i** (26.46%, IC_50_ = 18.53 ± 1.69 μM) suggesting that the *o*–CH_3_ group is essential for inhibiting activity of MAO. In addition to the *o*–CH_3_ group, compounds that included an electron–withdrawing group significantly enhanced MAO antagonism. Moreover, the contribution of different electron–withdrawing groups on the benzothiazole ring influenced activity differently. (1) Br > Cl > NO_2_ > F (**4d** (43.37%, IC_50_ = 64.83 ± 4.20 μM) > **4b** (39.27%, IC_50_ = 38.82 ± 3.76 μM) > **4n** (12.51%, IC_50_ > 150 μM) > **4m** (9.76%, IC_50_ > 150 μM)). (2) At the same time, substituting the halogen on the benzothiazole ring influenced the MAO inhibitory activity significantly. The *ortho–* or *meta–* position > *para–*position (Cl substituent **4b** > **4c** (21.59%, IC_50_ = 45.90 ± 2.66 μM), Br substituent **4d** > **4f** (24.15%, IC_50_ = 41.78 ± 2.94 μM)). Thus, substituents at the *para*–position reduced the inhibitory effect of compounds toward MAO. (3) Comparing the activities of *mono*– and *di*–substituted compounds with Cl or Br revealed that the *di*–Cl_2_ or *di*–Br_2_ substitution is not conducive to inhibiting MAO (Cl substituent **4b** > **4o** (10.19%, IC_50_ = 67.80 ± 1.26 μM), Br substituent **4d** > **4p** (19.80%, IC_50_ = 59.34 ± 3.22 μM)). According to the above–mentioned SAR results, we selected seven active target compounds to study the alternative inhibition of MAO–A and MAO–B further.

#### 2.2.2. Selective Inhibition of MAO–A and MAO–B by the Derivatives

The inhibitory activities of derivatives (**4a**–**4p**) against monoamine oxidase A (MAO–A) and MAO–B were evaluated with pagiline and clogiline as positive controls. The results are listed in Table 2. The six target compounds had negligible inhibitory activity against MAO–A. In contrast, the analogs all exhibited varying degrees of MAO–B antagonism. Among the six compounds, **4d** had the highest inhibitory effect on MAO–A, but only an inhibition rate of 18.05%. *Thus, this series of compounds may not have an antidepressant activity or exert antidepressant activity by other mechanisms. The specific results need to be further verified and confirmed by* in vivo *experiments.* In particular, **4i** (61.17%, IC_50_ = 16.49 ± 3.59 μM) displayed the strongest MAO–B inhibitory activity. Inhibition of MAO–B by **4g** (IC_50_ = 12.12 ± 3.47 μM) was more significant, with the inhibition rate reaching 51.39%. However, the inhibition rate of **4g** was not as strong as the positive control pargyline.

SAR data of benzothiazole–isoquinoline derivatives toward MAO–B are shown in Table 2. Based on the most active compound **4i**, the introduction of the *p*–CH_3_ group on the benzothiazole ring is beneficial for inhibiting MAO–B, and the inhibitory activity of MAO–B is also very significant. Introducing the –CH_3_ at the *para*–position yielded stronger inhibition than placing this moiety at the *ortho*–position. The introduction of different electron–withdrawing groups had different effects on the MAO–B antagonism of target compounds: (1) Cl > Br, **4c** (61.17%, IC_50_ = 9.13 ± 4.17 μM) > **4f** (61.17%, IC_50_ = 47.61 ± 1.82 μM). (2) At the same time, substitution of the halogen on the benzothiazole ring also greatly affected the MAO–B inhibitory activity. With the *para*–position > *meta*– or *ortho*–position, (Cl substituent **4c** (48.45%, IC_50_ = 9.13 ± 4.17 μM) > **4b** (39.15%, IC_50_ = 40.84 ± 2.67 μM); Br substituent **4f** (46.49%, IC_50_ = 47.61 ± 1.82 μM) > **4d** (46.00%, IC_50_ = 3.26 ± 2.78 μM)). In addition, the benzothiazole–isoquinoline derivatives exhibited remarkable selectivity, demonstrating that they are MAO–B inhibitors with high potency and selectivity.

#### 2.2.3. ChE Inhibition by the Benzothiazole–Isoquinoline Derivatives

The inhibitory activities of all benzothiazole–isoquinoline derivatives (**4a**–**4p**) against BuChE were assessed in vitro by the modified Ellman method [57], and the positive control was tacolin. The results showed that a series of benzothiazole–isoquinoline derivatives showed no inhibitory effect on AChE. The majority of the benzothiazole–isoquinoline compounds demonstrated significant BuChE inhibitory activity because of the intricate design of the pharmacophore fusion strategy. Among them, compound **4d** with *o*–Br as the substituent had the highest activity, and the inhibition rate was 77.76%; however, its inhibitory activity was far weaker than that of the positive control tacolin (99.41%, IC_50_ = 14.61 ± 5.81 μM).

The SAR results of benzothiazole–isoquinoline derivatives on BuChE are listed in Table 3. Different substitution positions displayed different anti–BuChE activities when the benzene ring was an electron–withdrawing substituent. The determined activity sequence was *o*–Br > *m*–Cl > *m*–Br > *p*–Cl > 2,6–Cl_2_ > 2,6–Br_2_ > *o*–Cl > *p*–Br > *p*–F > *p*–NO_2_. Compound **4d** with *o*–Br as the substituent was observed to have the highest activity. The effect of the substitution position on activity differs when the benzene ring is an electron–donating substituent. The determined activity sequence was *o*–CH_3_ > *m*–OCH_3_ > *o*–OCH_3_ > *p*–CH_3_ > *p*–OCH_3_ > *m*–CH_3_. A comparison of the abilities of *mono–* and *di*–substituted compounds with Cl or Br was conducted: *di*–Cl_2_ or *di*–Br_2_ substitution was not conducive to inhibiting BuChE. (Cl substituent **4b** (69.44%, IC_50_ = 17.59 ± 1.78 μM) > **4o** (52.16%, IC_50_ = 21.76 ± 2.85 μM); Br substituent **4d** (77.76%, IC_50_ = 14.61 ± 5.81 μM) > **4p** (43.84%, IC_50_ = 27.12 ± 1.73 μM)).

### 2.3. In Vivo Biological Evaluation

The antidepressant activities of fluoxetine and benzothiazole–isoquinoline derivatives, as measured by the immobility time in the FST, are listed in Table 4. *Most benzothiazole–isoquinoline derivatives induced a considerable reduction in the immobility time, suggesting that the series of derivatives may have antidepressant activity, but not through the monoamine oxidase mechanism.* Specifically, **4d** and **4g** exhibited the strongest antidepressant effects, resulting in a considerable reduction in immobility duration in comparison to the control group (*p* < 0.001).

Compounds **4d** and **4g** reduced the immobility duration and displayed stronger %DID (percentage decrease in immobility duration) values than the other examine derivatives. The %DID values of **4d** (65.06%) and **4g** (61.32%) were slightly lower than those of fluoxetine (82.2%) at a concentration of 30 mg/kg in the FST. However, compared with the control group, all seven derivatives had significant antidepressant effects and their %DIDs were all greater than 50%.

### 2.4. Cellular Toxicity

Six benzothiazole–isoquinoline derivatives (**4b**, **4c**, **4d**, **4f**, **4g**, **4i**) with strong activity were tested for cytotoxicity. The results of the MTT experiment are presented in Figure 2. Compared with the blank control group, the test results revealed that at 0–100 μM, the viability of the six benzothiazole–isoquinoline derivatives on cells all reached more than 90% and only showed weak toxicity at high concentrations.

Through the Acridine Orange (AO) fluorescence staining experiment, compared with the blank control group, the test results revealed that at 100 μM, the six benzothiazole–isoquinoline derivatives had no significant difference in cell viability, and the results are shown in Appendix A.

### 2.5. Molecular Docking

Molecular modeling was performed using the Vina (Simina) docking program to further investigate the interaction mode of compound **4g** with BuChE and MAO–B (Figure 3). The interaction modes of compound **4g** with BuChE and MAO–B revealed that derivative **4g** binds to the active sites of BuChE and MAO–B, occupying the entire enzymatic CAS, mid–gorge site and PAS [58,59]. The docking results showed that **4g** produced favorable binding energy values for both enzymes, with the greatest result observed for BuChE (−11.1 kcal·mol^−1^) and MAO–B (−9.2 kcal·mol^−1^). The blood–brain barrier (BBB) value for **4g** was 0.84 suggesting the relatively good passage of this compound across the BBB. Upon docking with BuChE, the tetrahydroisoquinoline moiety binds to PAS, forming a π–π stacking interaction with Trp82 and Tyr332, whereas the amide moiety forms a hydrogen bond with Thr120. Upon docking with MAO–B, the amide moieties form hydrogen bonds with Gln206 and Ile199. These results indicate that the derivatives are dual inhibitors of BuChE and MAO–B, which is consistent with the results obtained from previous analyses.

## 3. Experimental Protocols

Chemical reagents were obtained and used directly from standard suppliers. Use Agilent liquid chromatograph to detect its purity. Column temperature: 35 °C, detection wavelength: 204 nm, mobile phase: methanol: water (V:V, 82:18), chromatographic column: DIAMONSIL^TM^ C_18_ column (250 mm × 4.6 mm, 5 μm).

### 3.1. The Synthetic Routes of Novel Benzothiazole–Isoquinoline Derivatives

#### 3.1.1. General Procedure for the Synthesis of Substituted Benzo[d]thiazol–2–amine (**2a**–**2p**)

In total, 50 mmol of substituted aniline (**1a**–**1p**) and 250 mmol of KSCN were added to about 15 mL of acetic acid. Under ice bath conditions, glacial acetic acid mixed with 75 mmol of liquid bromine was slowly added dropwise, then at room temperature (RT) for 4 h. The reaction solution was poured into ice water to adjust pH. After the solid was precipitated, suction was filtered, the solid was dried, and recrystallized with ethanol to obtain the pure substituted benzo[d]thiazol–2–amine(**2a**–**2p**).

#### 3.1.2. General Procedure for the Synthesis of Substituted *N*–(benzo[d]thiazol–2–yl)–2–chloroacetamide (**3a**–**3p**)

Dissolved 50 mmol of substituted benzo[d]thiazol–2–amine (**2a**–**2p**) in tetrahydrofuran (THF), added 75 mmol of triethylamine, slowly added chloroacetyl chloride dropwise under ice bath conditions and stirred at RT for 1 h. To remove trimethylamine hydrochloride, the solvent was evaporated at decreased pressure and the residue was washed with water. The residue was dried and recrystallized to give the product substituted N–(benzo[d]thiazol–2–yl)–2–chloroacetamide (**3a**–**3p**).

#### 3.1.3. General Procedure for the Synthesis of Substituted (*R*)–*N*–(benzo[d]thiazol–2–yl)–2–(1–phenyl–3,4–dihydroisoquinolin–2(1*H*)–yl)acetamide (**4a**–**4p**)

Dissolved 50 mmol of substituted N–(benzo[d]thiazol–2–yl)–2–chloroacetamide (**3a**–**3p**) in DMF, added 75 mmol of triethylamine and slowly added 75 mmol of (*S*)–1–phenyl–1,2,3,4–tetrahydroisoquinoline stirred in an oil bath under reflux for 1.5–2 h at 80 °C. The reaction solution was poured into ice water to adjust pH. After, the solid was precipitated, suction filtered and dried, and recrystallized to obtain pure (*R*)–N–(benzo[d]thiazol–2–yl)–2–(1–phenyl–3,4–dihydroisoquinolin–2(1*H*)–yl)acetamide (**4a**–**4p**).

### 3.2. The Spectral Information of Novel Benzothiazole–Isoquinoline Derivatives

#### 3.2.1. (*R*)–*N*–(4–chlorobenzo[d]thiazol–2–yl)–2–(1–phenyl–3,4–dihydroisoquinolin–2(1*H*)–yl)acetamide (**4a**)

HPLC/Purity: 91.6% (t_R_ = 9.089), yield: 82.63%. mp: 245.6–247.0 °C.IR (KBr) cm^−1^: 3288, 1619, 1520, 1225.^1^H NMR(CDCl_3,_ 300 MHz): *δ* 9.98 (1H, s, –NH), 7.33–7.12 (12H, m, –C_6_H_5_), 5.69 (1H, s, –CH), 4.86 (2H, t, –CH_2_–), 3.58–3.49 (4H, m, –C_6_H_5_), ^13^C NMR (CDCl_3_, 75 MHz):*δ* 139.50, 138.98, 138.88, 138.67, 137.92, 137.74, 137.06, 136.38, 136.17, 135.86, 71.83, 68.46, 59.01, 39.01.ESI–MS calcd for C_24_H_21_ClN_3_OS^+^([M+H]^+^): 433.1097; found: 433.1078.

#### 3.2.2. (*R*)–*N*–(5–chlorobenzo[d]thiazol–2–yl)–2–(1–phenyl–3,4–dihydroisoquinolin–2(1*H*)–yl)acetamide (**4b**)

HPLC/Purity: 100.0% (t_R_ = 10.829), yield: 81.48%. mp: 248.1–250.9 °C.IR (KBr) cm^−1^: 3287, 1617, 1515, 1228.^1^H NMR(CDCl_3,_ 300 MHz): *δ* 9.18 (1H, s, –NH), 7.58–6.68 (12H, m, –C_6_H_5_), 5.28 (1H, s, –CH), 4.72 (2H, t, –CH_2_), 3.36–3.30 (4H, m, –C_6_H_5_), ^13^C NMR (CDCl_3_, 75 MHz):*δ* 169.03, 142.60, 138.78, 137.39, 134.70, 133.62, 130.04, 129.55, 128.95, 128.82, 128.57, 128.10, 126.59, 126.22, 124.17, 119.38, 117.28, 77.30, 77.05, 76.79, 68.56, 58.95, 49.58, 29.40. ESI–MS calcd for C_24_H_21_ClN_3_OS^+^([M+H]^+^): 433.1097; found: 433.1078.

#### 3.2.3. (*R*)–*N*–(6–chlorobenzo[d]thiazol–2–yl)–2–(1–phenyl–3,4–dihydroisoquinolin–2(1*H*)–yl)acetamide (**4c**)

HPLC/Purity: 92.0% (t_R_ = 11.526), yield: 80.17%. mp: 240.7–243.1 °C.IR (KBr) cm^−1^: 3280, 1615, 1513, 1224.^1^H NMR(CDCl_3,_ 300 MHz): *δ* 12.11 (1H, s, –NH), 7.71–7.01 (12H, m, –C_6_H_5_), 4.98 (1H, s, –CH), 4.72 (2H, t, –CH_2_–), 2.98–2.84 (4H, m, –C_6_H_5_), ^13^C NMR (CDCl_3_, 75 MHz):*δ* 170.83, 158.79, 147.88, 143.75, 138.33, 134.75, 133.70, 129.93, 129.02, 128.14, 127.89, 126.45, 126.11, 66.52, 57.04, 48.29, 29.24. ESI–MS calcd for C_24_H_21_ClN_3_OS^+^([M+H]^+^): 433.1097; found: 433.1078.

#### 3.2.4. (*R*)–*N*–(4–bromobenzo[d]thiazol–2–yl)–2–(1–phenyl–3,4–dihydroisoquinolin–2(1*H*)–yl)acetamide (**4d**)

HPLC/Purity: 100.0% (t_R_ = 18.273), yield: 82.66%. mp: 254.3–255.9 °C.IR (KBr) cm^−1^: 3282, 1619, 1515, 1223.^1^H NMR(CDCl_3,_ 300 MHz): *δ* 7.65 (1H, s, –NH), 7.64–6.61 (12H, m, –C_6_H_5_), 4.72 (1H, s, –CH), 4.42 (2H, t, –CH_2_–), 3.65–2.72 (4H, m, –C_6_H_5_), ^13^C NMR (CDCl_3_, 75 MHz):*δ* 171.11, 146.46, 142.72, 136.91, 129.47, 129.02, 128.83, 128.77, 128.63, 128.51, 128.29, 126.32, 126.19, 126.02, 125. 95, 125.66, 116.22, 77.31, 77.26, 77.05, 76.80, 48.02, 43.48, 29.73, 17.57. ESI–MS calcd for C_24_H_21_BrN_3_OS^+^([M+H]^+^): 477.0573; found: 477.0578.

#### 3.2.5. (*R*)–*N*–(5–bromobenzo[d]thiazol–2–yl)–2–(1–phenyl–3,4–dihydroisoquinolin–2(1*H*)–yl)acetamide (**4e**)

HPLC/Purity: 100.0% (t_R_ = 6.034), yield: 79.03%. mp: 255.3–256.9 °C.IR (KBr) cm^−1^: 3280, 1611, 1511, 1227.^1^H NMR(CDCl_3,_ 300 MHz): *δ* 9.30 (1H, s, –NH), 7.97–6.70 (12H, m, –C_6_H_5_), 6.69 (1H, s, –CH), 4.73 (2H, t, –CH_2_–), 3.38–3.14 (4H, m, –C_6_H_5_), ^13^C NMR (CDCl_3_, 75 MHz):*δ* 169.35, 142.47, 139.63, 137.16, 133.50, 131.30, 129.52, 128.95, 128.87, 128.58, 128.20, 126.69, 126.30, 123.81, 120.29, 119.44, 109.89, 77.29, 77.24, 77.03, 76.78, 68.63, 59.04, 49.67, 29.35. ESI–MS calcd for C_24_H_21_BrN_3_OS^+^([M+H]^+^): 477.0573; found: 477.0578.

#### 3.2.6. (*R*)–*N*–(6–bromobenzo[d]thiazol–2–yl)–2–(1–phenyl–3,4–dihydroisoquinolin–2(1*H*)–yl)acetamide (**4f**)

HPLC/Purity: 93.5% (t_R_ = 19.415), yield: 77.40%. mp: 252.6–254.3 °C.IR (KBr) cm^−1^: 3289, 1611, 1517, 1220.^1^H NMR(CDCl_3,_ 300 MHz): *δ* 10.35 (1H, s, –NH), 7.91–7.15 (12H, m, –C_6_H_5_), 6.67 (1H, s, –CH), 4.73 (2H, t, –CH_2_–), 3.48–2.86 (4H, m, –C_6_H_5_), ^13^C NMR (CDCl_3_, 75 MHz):*δ* 169.79, 157.28, 147.51, 142.03, 137.25, 133.46, 129.72, 129.67, 128.93, 128.81, 128.53, 128.24, 126.66, 126.18, 123.93, 122.29, 116.94, 77.27, 77.02, 76.76, 69.03, 58.24, 50.37, 29.38. ESI–MS calcd for C_24_H_21_BrN_3_OS^+^([M+H]^+^): 477.0573; found: 477.0578.

#### 3.2.7. (*R*)–*N*–(4–methylbenzo[d]thiazol–2–yl)–2–(1–phenyl–3,4–dihydroisoquinolin–2(1*H*)–yl)acetamide (**4g**)

HPLC/Purity: 93.5% (t_R_ = 10.537), yield: 80.78%. mp: 247.6–249.7 °C.IR (KBr) cm^−1^: 3287, 1612, 1514, 1229.^1^H NMR(CDCl_3,_ 300 MHz): *δ* 9.28 (1H, s, –NH), 7.37–7.02 (12H, m, –C_6_H_5_), 6.68 (1H, s, –CH), 4.71 (2H, t, –CH_2_–), 3.47–2.82 (4H, m, –C_6_H_5_), 1.56 (3H, s, –CH_3_) ^13^C NMR (CDCl_3_, 75 MHz):*δ* 169.04, 142.77, 137.80, 137.49, 133.39, 132.82, 130.04, 129.56, 128.93, 128.86, 128.52, 128.40, 128.19, 126.56, 126.27, 121.60, 117.69, 77.27, 77.02, 76.77, 68.69, 59.20, 50.13, 29.70, 17.80. ESI–MS calcd for C_25_H_24_N_3_OS^+^([M+H]^+^): 413.1673; found: 413.1678.

#### 3.2.8. (*R*)–*N*–(5–methylbenzo[d]thiazol–2–yl)–2–(1–phenyl–3,4–dihydroisoquinolin–2(1*H*)–yl)acetamide (**4h**)

HPLC/Purity: 98.5% (t_R_ = 9.147), yield: 78.27%. mp: 241.3–243.6 °C.IR (KBr) cm^−1^: 3290, 1617, 1514, 1226.^1^H NMR(CDCl_3,_ 300 MHz): *δ* 9.26 (1H, s, –NH), 7.54–7.04 (12H, m, –C_6_H_5_), 6.79 (1H, s, –CH), 4.73 (2H, t, –CH_2_–), 3.37–3.15 (4H, m, –C_6_H_5_), 1.56 (3H, s, –CH_3_),^13^C NMR (CDCl_3_, 75 MHz):*δ* 169.29, 142.52, 141.63, 139.90, 137.30, 134.23, 133.57, 129.53, 129.96, 128.82, 128.57, 128,12, 126.62, 126.25, 121.64, 118.17, 117.26, 110.72, 77.27, 77.02, 76.76, 68.56, 59.01, 49.60, 29.37, 20.83. ESI–MS calcd for C_25_H_24_N_3_OS^+^([M+H]^+^): 413.1673; found: 413.1678.

#### 3.2.9. (*R*)–*N*–(6–methylbenzo[d]thiazol–2–yl)–2–(1–phenyl–3,4–dihydroisoquinolin–2(1*H*)–yl)acetamide (**4i**)

HPLC/Purity: 95.1% (t_R_ = 15.277), yield: 76.96%. mp: 244.8–246.9 °C.IR (KBr) cm^−1^: 3288, 1612, 1519, 1220.^1^H NMR(CDCl_3,_ 300 MHz): *δ* 10.35 (1H, s, –NH), 7.70–7.04 (12H, m, –C_6_H_5_), 6.69 (1H, s, –CH), 4.79 (2H, t, –CH_2_–), 3.28–2.84 (4H, m, –C_6_H_5_), 1.66 (3H, s, –CH_3_), ^13^C NMR (CDCl_3_, 75 MHz):*δ* 169.59, 159.23, 146.38, 142.10, 137.33, 134.00, 133.53, 132.37, 129.73, 129.54, 129.50, 128.88, 128.78, 128.74, 128.52, 128.17, 127.72, 126.59, 126.10, 121.19, 120.59, 119.40, 77.45, 77.02, 76.60, 68.99, 58.26, 50.31, 29.36, 21.47. ESI–MS calcd for C_25_H_24_N_3_OS^+^([M+H]^+^): 413.1673; found: 413.1678.

#### 3.2.10. (*R*)–*N*–(4–methoxybenzo[d]thiazol–2–yl)–2–(1–phenyl–3,4–dihydroisoquinolin–2(1*H*)–yl)acetamide (**4j**)

HPLC/Purity: 93.1% (t_R_ = 10.766), yield: 75.88%. mp: 253.7–254.9 °C.IR (KBr) cm^−1^: 3281, 1610, 1516, 1222.^1^H NMR(CDCl_3,_ 300 MHz): *δ* 9.91 (1H, s, –NH), 7.76–6.69 (12H, m, –C_6_H_5_), 4.72 (1H, s, –CH), 3.95 (2H, t, –CH_2_–), 3.37–2.82 (4H, m, –C_6_H_5_), 1.26 (3H, s, –CH_3_), ^13^C NMR (CDCl_3_, 75 MHz):*δ* 169.31, 149.08, 142.86, 137.70, 133.77, 129.48, 128.92, 128.70, 128.51, 127.97, 126.41, 126.12, 124.64, 120.27, 117.10, 112.69, 111.09, 77.29, 77.04, 76.78, 68.49, 59.03, 56.25, 49.75, 29.52. ESI–MS calcd for C_25_H_24_N_3_O_2_S^+^([M+H]^+^): 429.1573; found: 429.1578.

#### 3.2.11. (*R*)–*N*–(5–methoxybenzo[d]thiazol–2–yl)–2–(1–phenyl–3,4–dihydroisoquinolin–2(1*H*)–yl)acetamide (**4k**)

HPLC/Purity: 91.4% (t_R_ = 8.686), yield: 73.37%. mp: 248.9–250.1 °C.IR (KBr) cm^−1^: 3288, 1615, 1512, 1229.^1^H NMR(CDCl_3,_ 300 MHz): *δ* 9.91 (1H, s, –NH), 7.31–6.71 (12H, m, –C_6_H_5_), 5.29 (1H, s, –CH), 4.72 (2H, t, –CH_2_–), 3.95–3.08 (4H, m, –C_6_H_5_), 2.80 (3H, s, –CH_3_), ^13^C NMR (CDCl_3_, 75 MHz):*δ* 160.24, 138.88, 133.71, 129.71, 129.55, 128.96, 128.78, 128.57, 128.03, 126.54, 126.17, 111.50, 110.00, 105.07, 77.28, 77.03, 76.78, 68.49, 58.95, 55.35, 49.48, 29.37. ESI–MS calcd for C_25_H_24_N_3_O_2_S^+^([M+H]^+^): 429.1573; found: 429.1578.

#### 3.2.12. (*R*)–*N*–(6–methoxybenzo[d]thiazol–2–yl)–2–(1–phenyl–3,4–dihydroisoquinolin–2(1*H*)–yl)acetamide (**4l**)

HPLC/Purity: 93.3% (t_R_ = 13.066), yield: 75.24%. mp: 247.9–249.4 °C.IR (KBr) cm^−1^: 3286, 1617, 1514, 1229.^1^H NMR(CDCl_3,_ 300 MHz): *δ* 9.15 (1H, s, –NH), 7.33–7.16 (12H, m, –C_6_H_5_), 6.69 (1H, s, –CH), 4.73 (2H, t, –CH_2_–), 3.80–3.09 (4H, m, –C_6_H_5_), 1.58 (3H, s, –CH_3_), ^13^C NMR (CDCl_3_, 75 MHz):*δ* 169.44, 156.87, 142.70, 142.11, 137.33, 133.53, 133.50, 129.73, 128.89, 128.80, 128.53, 128.18, 126.60, 126.12, 121.63, 115.23, 104.31, 77.30, 77.25, 77.04, 76.79, 68.99, 58.22, 55.84, 50.29, 29.37, 14.13. ESI–MS calcd for C_25_H_24_N_3_O_2_S^+^([M+H]^+^): 429.1573; found: 429.1578.

#### 3.2.13. (*R*)–*N*–(6–fluorobenzo[d]thiazol–2–yl)–2–(1–phenyl–3,4–dihydroisoquinolin–2(1*H*)–yl)acetamide (**4m**)

HPLC/Purity: 92.0% (t_R_ = 11.526), yield: 86.40%. mp: 255.6–258.1 °C.IR (KBr) cm^−1^: 3280, 1616, 1513, 1227.^1^H NMR(CDCl_3,_ 300 MHz): *δ* 11.98 (1H, s, –NH), 7.83–6.95 (12H, m, –C_6_H_5_), 4.92 (1H, s, –CH), 3.37 (2H, t, –CH_2_–), 2.92–2.44 (4H, m, –C_6_H_5_), ^13^C NMR (CDCl_3_, 75 MHz):*δ* 157.64, 129.40, 128.51, 128.18, 127.37, 125.76, 121.63, 114.22, 108.17, 66.00, 56.49, 47.77, 28.72.ESI–MS calcd for C_24_H_21_FN_3_OS^+^([M+H]^+^): 417.1373; found: 417.1378.

#### 3.2.14. (*R*)–*N*–(6–nitrobenzo[d]thiazol–2–yl)–2–(1–phenyl–3,4–dihydroisoquinolin–2(1*H*)–yl)acetamide (**4n**)

HPLC/Purity:90.8% (t_R_ = 6.333), yield: 82.59%. mp: 240.6–243.2 °C.IR (KBr) cm^−1^: 3283, 1610, 1517, 1223.^1^H NMR(CDCl_3,_ 300 MHz): *δ* 12.43 (1H, s, –NH), 7.84–6.98 (12H, m, –C_6_H_5_), 6.62 (1H, s, –CH), 4.96 (2H, t, –CH_2_–), 3.15–2.84 (4H, m, –C_6_H_5_), ^13^C NMR (CDCl_3_, 75 MHz):*δ* 170.87, 162.99, 153.39, 143.09 (d, *J* = 29.3 Hz), 137.80, 134.23, 129.41, 128.50, 128.37, 128.27, 127.38, 125.94, 125.60, 121.76, 120.57, 119.03, 66.00, 56.60, 47.79, 28.73.ESI–MS calcd for C_24_H_21_N_4_O_3_S^+^([M+H]^+^): 444.1373; found: 444.1378.

#### 3.2.15. (*R*)–*N*–(4,6–dichlorobenzo[d]thiazol–2–yl)–2–(1–phenyl–3,4–dihydroisoquinolin–2(1*H*)–yl)acetamide (**4o**)

HPLC/Purity: 94.8% (t_R_ = 22.278), yield: 71.83%. mp: 253.5–256.1 °C.IR (KBr) cm^−1^: 3287, 1614, 1516, 1223.^1^H NMR(CDCl_3,_ 300 MHz): *δ* 9.87 (1H, s, –NH), 8.42–6.67 (12H, m, –C_6_H_5_), 5.29 (1H, s, –CH), 4.70 (2H, t, –CH_2_–), 3.45–2.84 (4H, m, –C_6_H_5_), ^13^C NMR (CDCl_3_, 75 MHz):*δ* 169.24, 133.63, 133.46, 129.61, 128.85, 128.82, 128.72, 128.51, 128.06, 127.88, 126.41, 126.11, 121.33, 77.28, 77.23, 77.02, 76.77, 68.78, 59.17, 50.33, 29.70. ESI–MS calcd for C_24_H_21_Cl_2_N_3_OS^+^([M+H]^+^): 467.0673; found: 467.0678.

#### 3.2.16. (*R*)–*N*–(4,6–dibromobenzo[d]thiazol–2–yl)–2–(1–phenyl–3,4–dihydroisoquinolin–2(1*H*)–yl)acetamide (**4p**)

HPLC/Purity: 93.0% (t_R_ = 10.816), yield: 73.12%. mp: 258.6–260.1 °C.IR (KBr) cm^−1^: 3280, 1612, 1510, 1221.^1^H NMR(CDCl_3,_ 300 MHz): *δ* 9.86 (1H, s, –NH), 7.91–7.03 (12H, m, –C_6_H_5_), 6.60 (1H, s, –CH), 4.83 (2H, t, –CH_2_–), 3.25–2.78 (4H, m, –C_6_H_5_), ^13^C NMR (CDCl_3_, 75 MHz):*δ* 178.73, 152.94, 147.56, 144.13, 143.76, 141.22, 139.45, 138.36, 137.56, 135.96, 135.68, 132.14, 59.14, 38.94. ESI–MS calcd for C_24_H_21_Br_2_N_3_OS^+^([M+H]^+^): 554.9673; found: 554.9678.

### 3.3. Determination of the Inhibitory Activity of **4a**–**4p** on MAO by Holts Method

#### 3.3.1. Preparation of MAO

Male SD rats were sacrificed by decapitation, operated on ice, their livers were removed, and the liver surface was washed with precooled phosphate buffer. The liver was cut into pieces, and after washing (using PBS), 20 mL of 0.3 M sucrose buffer was added for homogenization. After fully homogenized, it was balanced by a tray balance. Low–temperature differential centrifugation was performed, and the obtained SD rat liver mitochondrial concentrate was the experimental monoamine oxidase stock solution, and was stored in a −80 °C refrigerator.

#### 3.3.2. MAO Inhibitory Activity and IC_50_ Detection

Added 25 µL of enzyme dilution solution and 25 µL of test derivatives (control group plus rasagiline, blank group plus 10% DMSO) to 96–well plate, incubated at 37 °C for 1 h, then 120 µL of substrate was added, 80 µL of chromogenic solution was added, incubated at 37 °C for 30 min, and its absorbance measured at 490 nm. In addition, in order to exclude the influence of protein, a protein control group was established, only the enzyme dilution was added, and the rest were replaced with PBS, and the absorbance value was detected at a wavelength of 490 nm. The inhibition rate of each concentration was substituted into the calculation software GraphPad Prism 8 to calculate the IC_50_ value.

#### 3.3.3. MAO–A Inhibitory Activity and IC_50_ Detection

Added 25 µL of enzyme solution and 25 µL of 10 μM pargyline solution to the 96–well plate in sequence, and incubated at 37 °C for 30 min, other operations were the same as 4.3.2 Monoamine Oxidase Inhibitory Activity and IC_50_ Detection. The inhibition rate of each concentration was substituted into the calculation software GraphPad Prism 8 to calculate the IC_50_ value.

#### 3.3.4. MAO–B Inhibitory Activity and IC_50_ Detection

Added 25 µL of enzyme solution and 25 µL of 10 μM chlorgiline solution to the 96–well plate in sequence, and incubated at 37 °C for 30 min, other operations were the same as 4.3.2 Monoamine Oxidase Inhibitory Activity and IC_50_ Detection. The inhibition rate of each concentration was substituted into the calculation software GraphPad Prism 8 to calculate the IC_50_ value.

### 3.4. Determination of the Inhibitory Activity of Target Compounds **4a**–**4p** on ChE

Compounds **4a**–**4p** were dissolved in DMSO and then diluted in 10% DMSO to obtain sample solutions of different concentrations. Each concentration was tested three times in parallel, and 96 empty plates were taken, and 20 µL of the test sample or positive drug (10% DMSO was added to the blank control), 40 µL of AChE or BuChE (0.2 U/mL) and 100 µL of DTNB (0.001 mol/L) were added in turn, mixed well and incubated in a constant temperature water bath at 37 °C for 15 min. Then, 20 µL of the substrate ATCI or BTCI (0.001 mol/L) was added, and after mixing evenly, the mixture was incubated in a constant temperature water bath at 37 °C for 3 min, and the absorbance value was measured at 412 nm.

### 3.5. Forced Swimming Test

Pharmacological experiments of antidepressant activity used FST in mice, fluoxetine hydrochloride was used as a positive control, and the vehicle solvent PFG–400 was used as a blank control. The experiment used ICR male mice (20 ± 2 g). Six benzothiazole–quinoline derivatives were dissolved in polyethylene glycol–400, and 30 min after administration (i.p., 30 mg/kg), the mice were each placed in a glass cylinder, recording the time of immobility. Duration of immobility outcomes is expressed as mean ± SEM. The *t*-test was used to make comparisons between groups.

### 3.6. Cell Culture

L929 (Shanghai Cell Bank, Chinese Academy of Sciences) cells were cultured in 10% fetal bovine serum: penicillin–streptomycin solution: MEM mixed medium with serum–free medium (10:1:90), and subcultured in a 5% carbon dioxide cell incubator at 37 °C, and the medium changed every other day.

### 3.7. Assessment of Cytotoxicity

In this experiment, L929 cells in the logarithmic growth phase were selected, and the cell density was adjusted to 2 × 10^4^ cells per well. The cells were seeded in 96–well plates, and the 96–well plates seeded with cells were cultured in a 37 °C 5% carbon dioxide cell incubator. After 24 h, different concentrations of the test compound medium (6.5, 12.5, 25, 50, 100 μM) were added, and the blank group was added with drug–free medium, shaken gently, and placed at 37 °C 5%. The cells were cultured in a carbon dioxide cell incubator for 24 h, and each concentration was replicated 6 times. After 24 h, 20 μL of 5 mg/kg MTT solution (prepared in PBS with pH = 7.6) was added in the dark, and cultured in the cell incubator for another 4 h. After 4 h, the 96–well plate was taken out and the original medium was discarded. Added 160 μL DMSO, put it on a shaker and mixed well for about 10 min, used a microplate reader to measure its absorbance at 490 nm, recorded the results and analyzed.

### 3.8. Analyze Cell Viability by AO Staining

L929 cells in the logarithmic growth phase were selected, and the cell density was adjusted to 8 × 10^5^ per well with a cell counter to inoculate in a 6–well plate, and the 6–well plate inoculated with cells was cultured in a 37 °C 5% carbon dioxide cell incubator. After 24 h, 1 mL of medium containing 100 μM of the compound to be tested was added (medium without drug was added to the blank group), shaken gently, and cultured in a 37 °C 5% carbon dioxide cell incubator. After 8 h, the medium was discarded, PBS was added to wash once, 1 mL of PBS and 80 μL of AO solution were slowly added in the dark, mixed for 5 min to make the staining uniform and sufficient, after 5 min, the stain was aspirated and washed twice with PBS, and observed under a fluorescence microscope, the results saved and analyzed.

### 3.9. Molecular Docking

Molecular simulations and docking experiments were performed using the Vina (Simina) docking program. The protein crystal structures of human MAO–B and BuChE were obtained from the Protein Crystal Database (PDBID: 2Z5X and 4BDS), resolution is 5Å (10^−10^ m). It mainly simulates the docking process between inhibitors and MAO–B and BuChE. Compound **4g** was selected as the inhibitor target for molecular modeling, and the initial structures of MAO–B and BuChE protein crystals were processed with default parameters, the pocket is the FAD ligand center, molecular simulations were performed between the optimized protein crystal structure and compound **4g**. The inhibitory effect was evaluated at the molecular level, which provided ideas for the design of better MAO and BuChE inhibitors in the future.

## 4. Conclusions

In this report, we designed and synthesized a series of (*R*)–*N*–(benzo[d]thiazol–2–yl)–2–(1–phenyl–3,4–dihydroisoquinolin–2(1*H*)–yl)acetamides as new multi–target inhibitors that potentially display multi–functional anti–neurodegenerative and antidepressant activities. Substituents on the benzene ring of benzothiazole were changed to investigate their effects on the selective inhibitory activities against MAO–B and BuChE, and SAR analysis was carried out. The inhibitory effect of benzothiazole–isoquinoline derivatives on MAO–A was found to be substantially less than against MAO–B. Compound **4d** displayed a mild inhibitory effect toward MAO–A, whereas six derivatives reduced MAO–B activity significantly. Compounds **4g** and **4i** gave the strongest inhibitory activity toward MAO–B. All derivatives exhibited selective inhibition of BuChE, with **4b**–**4e** and **4g** displaying the strongest inhibition. The FST showed that this series of benzothiazole–isoquinoline derivatives (**4d** and **4g**) reduced the immobility time significantly. *This observation indicates that this series of compounds does have antidepressant potential, but not through inhibition of monoamine oxidase and choline, and the specific mechanism of action needs to be explored further.* Subsequently, their cytotoxicity was assessed by the MTT assay and AO staining, and the results showed that the compounds were not toxic to L929 cells at effective concentrations. Molecular docking studies were carried out for **4g** to define the possible binding mode of this compound. The primary interaction site between the compound and the two enzymes was revealed, and was consistent with the observed in vitro MAO–B and BuChE inhibitory activities of these compounds, thereby broadening our understanding of the requirements for the compounds to achieve high affinity. The above results suggest that compound **4g** with multi–targeting is a promising drug candidate for treating neurodegenerative diseases complicated by depression. Nonetheless, further research on the disease pathways and pathophysiology of neurodegenerative diseases complicated by depression is required to clarify the pathogenesis of the disease and in the development of preventive and therapeutic measures.

## Data Availability

The data presented in this study and associated additional data are available upon request.

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
