# Peer review of "Synthesis, Characterization and Biological Evaluation of Benzothiazole–Isoquinoline Derivative"

_molecules, 2022, doi:10.3390/molecules27249062_

Round 1

Reviewer 1 Report

1.     The title of the manuscript is not clear. It would be better if the authors use the word characterization in their title after the word synthesis.

2.     The abbreviation MTDLs does not match with its full name i.e., multi-target ligands as mentioned by the authors in the abstract section and in the whole manuscript as well.

3.     In this abstract, the authors should mention the name of parent compound from where a series of compound were synthesized.

4.     Reaction conditions in scheme are missing.

5.     What is the basic mechanism behind the MTT Assay? It should be well explained. Which standard was used and how the assay was validated before running the samples. How was the wavelength optimized?

6.     The authors performed only two enzymatic activities? These two activities do not confirm that the synthesized compounds exhibited the targeted anti-Alzheimer activity. It reflects that the authors already decided that they will obtain those compounds who will exhibit the anti-Alzheimer activity. Detailed clarification is required.

7.     Where is the standard in figure 2?

8.     Experimental conditions in the Docking model of compound are missing? A detailed description is mandatory for this technique.

9.     What are the limitations of your study?

10.  There are several grammatical mistakes and syntax errors. The whole manuscript needs critical revision to remove all the grammatical mistakes and syntax errors.

Author Response

  1. 手稿的标题不清楚。如果作者在单词综合之后在标题中使用“表征”一词会更好。

答:谢谢你的建议,我们修改了文章中的标题。

  1. 缩写MTDLs与其全名不匹配,即作者在摘要部分和整个手稿中提到的多靶点配体。

答:您指出的缩写在全文中已得到更正。

  1. 在本摘要中,作者应提及合成一系列化合物的母体化合物的名称。

答:感谢您的建议,我们添加了合成一系列化合物的母体化合物的名称。 

  1. 方案中缺少反应条件。

答:您指出的缩写已在方案中得到纠正。

  1. MTT检测背后的基本机制是什么?应该很好地解释一下。在运行样品之前,使用哪种标准品以及如何验证测定。如何优化波长?

答:Mosmann等报道了一种基于四唑盐MTT(3-(4,5-二甲基噻唑-2-基)-2,5-二苯基四唑溴化物)的快速比色分析方法,该方法检测活细胞,但不检测死细胞,产生的信号取决于细胞活化程度。由于MTT可以通过活细胞的线粒体脱氢酶还原为甲臜,二甲基亚砜可以将甲溶解在细胞中,并且用酶标仪在490nm波长处测量其吸光度值,因此MTT已被用于建立哺乳动物细胞存活和增殖的定量比色分析方法。因此,该方法可用于测量细胞毒性、增殖或活化。 

  1. 作者只进行了两次酶促活动?这两种活性并不能证实合成的化合物表现出靶向抗阿尔茨海默氏症活性。它反映了作者已经决定他们将获得那些将表现出抗阿尔茨海默氏症活性的化合物。需要详细澄清。

答:单胺氧化酶会催化伯胺、仲胺和叔胺(包括单胺神经递质)氧化成相应的亚胺。在催化氧化的同时,FAD被氢化并还原为FADH2。FAD辅因子通过将氧气转化为过氧化氢来再生FADH2来实现催化循环。MAO-B主要通过选择性氧化苯乙胺、苄胺等胺类物质,可有效治疗阿尔茨海默病。此外,如果抑制AD患者大脑中BuChE向BuCh的水解,突触中BuCh的含量将明显增加,神经传递机制将更加顺畅。因此,这两种酶活性可以证实合成的化合物显示出靶向抗阿尔茨海默病的活性。

  1. 图 2 中的标准在哪里?

答:谢谢你的建议,我们已经修改了图2。

  1. 化合物对接模型中的实验条件缺失?此技术必须提供详细说明。

答:对于分子对接部分,稿件已经修改。

  1. 您的研究有哪些局限性?

答:这一观察表明,这一系列化合物确实具有抗抑郁潜力,但不是通过抑制单胺氧化酶和胆碱,具体作用机制有待进一步探讨。

  1. 有几个语法错误和语法错误。整份手稿需要认真修改,以消除所有语法错误和语法错误。

答:您指出的缩写在全文中已得到更正。

Author Response

  1. Answer: Thank you for your suggestion, we have modified the figure 1.
  2. Answer: Compound yield is for all three step combinations.

  3. Answer: CAS means the catalytic active (or anionic) site, PAS means the peripheral anionic site. ChE accelerates the progression of AD by two possible mechanisms. The CAS of ChE is responsible for the degradation of choline. The PAS generates a stable complex with β-amyloid peptide and by this means expedite the oligomerization of Aβ peptides and aggregation of senile plaques. Through molecular docking simulation, it was found that the compound interacted with the active sites of two enzymes, indicate that the compound is dual inhibitor of BuChE and MAO-B. If you are interested in the interpretation of CAS and PAS, you can learn more about them in the following references.

    (Pourshojaei Y, Abiri A, Eskandari K, Haghighijoo Z, Edraki N, Asadipour A. Phenoxyethyl Piperidine/Morpholine Derivatives as PAS and CAS Inhibitors of Cholinesterases: Insights for Future Drug Design. Sci Rep. 2019 Dec 27;9(1):19855. doi: 10.1038/s41598-019-56463-2. )

  4. Answer: Thank you for your suggestion, the abbreviations you pointed out have been corrected.

  5. Answer: Use Agilent liquid chromatograph to detect its purity. Column temperature: 35 ℃, detection wavelength: 204 nm, mobile phase: methanol: water (V:V, 82:18), chromatographic column: DIAMONSILTM C18 column (250 mm×4.6 mm, 5 μm). And the details have been supplemented in the manuscript.

  6. Answer: In the pre experiment, the concentration of MAO protein was 26.03 mg/mL through BCA protein quantitative kit.

  7. Answer: Thank you for your suggestion, the inhibitory activity and IC50 were processed by Graphpad Prism 8 software, and the corresponding supplements and modifications were also made in the text.

  8. Thank you for your suggestion. The results of BBB permeameter are predicted by Vina (Simina) docking program.

  9. The grammar and spelling you pointed out have been corrected in the whole manuscript.

Round 2

Reviewer 1 Report

I suggest the authors to consider the following title:

Synthesis, Characterization and Biological Evaluation of Benzothiazole-isoquinoline Derivatives
 Some comments have not been addressed appropriately: For example:

What are the limitations of your study?

There are several grammatical mistakes and syntax errors. 

Author Response

Dear reviewer,

It is a pleasure to submit our revised manuscript. We have revised our manuscript according to the reviewer’ comments. The parts modified are underlined in the article.

I would like to thank you for your kind consideration to publish this paper.  

Thank you for your consideration.
Sincerely yours,

Li-Ping Guan

1. Answer: Thank you for your suggestion about the title. We have changed the title to “Synthesis, Characterization and Biological Evaluation of Benzothiazole-isoquinoline Derivatives”

2. What are the limitations of your study?

Answer: The derivatives in this paper have been proved to have antidepressant activity through the classical forced swimming experiment, but compared with the enzyme activity experiment, it was found that the mechanism of action is not through the anti monoamine oxidase and anticholinesterase pathway. Therefore, the limitation of this paper is that the specific mechanism of antidepressant action of the derivatives needs to be further explored.

3. There are several grammatical mistakes and syntax errors.

Answer: The abbreviations you pointed out have been corrected in the whole manuscript.

Reviewer 2 Report

The authors have addressed the comments of this reviewer and have incorporated the suggested changes. 

Author Response

Thank you very much for your review.